# Handover Triggering Prediction with the Two-Step XGBOOST Ensemble Algorithm for Conditional Handover in Non-Terrestrial Networks

**Eunsu Kim and Inwhee Joe ***

Department of Computer Science, Hanyang University, Seoul 04763, Republic of Korea; rmsid1312@hanyang.ac.kr
* Correspondence: iwjoe@hanyang.ac.kr

**Abstract:** A Non-Terrestrial Network (NTN) is a network system that enables service for areas where terrestrial networks cannot cover. An NTN provides communication services using flying objects such as UAVs, HAPs, and satellites. In the case of satellites, they move in Earth's orbit at a constant speed. Ground services from continuously moving satellites cause frequent handovers. In addition, frequent handovers may come as a load between User Equipment (UE) and the communication system, which leads to degradation of service quality. Unlike Terrestrial Networks (TN), communication services are provided to UEs at altitudes ranging from 20 km to 35,584 km, rather than from base stations close to the ground. Service at high altitudes is unreliable due to the measurement values that were previously used as quality indicators to operate terrestrial networks. Moreover, service at high altitudes demands long-distance communication, and propagation delay occurs from the long-distance communication. In the 3GPP Rel. 17 document, it is suggested that the above problems should be solved. This paper tries to solve the problem by proposing the two-step XGBOOST, a CART-based Gradient Boosting Model. Handover in TN uses measurement-based conditional handover (CHO), but the measured values in the NTN environment are not valid. Using this, the distance between the UE and the center of the cell and the elevation angle are used to construct a model that predicts the HO triggering time point. In order to overcome the propagation delay caused by communication at a high altitude, a model that predicts the distance and elevation angle between the UE and the center of the cell considering the propagation delay is proposed. The model is composed of two-step XGBOOST. The one-step model is a model in which the UE predicts the distance and elevation angle between cell centers after propagation delay at the time when satellite position information is transmitted to the UE. The two-step model predicts handover triggering occurrence based on the data predicted by the one-step result. As a result of the experiment, the model considering the propagation delay showed about 8% better performance on average than the model not considering the propagation delay, and the XGBOOST model achieved an average F1-score of 0.9891 in the propagation delay experiments.

**Keywords:** non-terrestrial network; conditional handover; machine learning; XGBOOST

## 1. Introduction

In existing networks, communication between User Equipments (UEs) is accomplished through terrestrial base stations. However, this communication service is limited to the coverage area of the installed base station, and it cannot be provided outside that region. To address these limitations, a communication method called Non-Terrestrial-Network (NTN) 3GPP was proposed. NTN utilizes satellites capable of providing communication services across all regions, enabling coverage in areas not served by ground-based base stations. NTN leverages satellites, High-Altitude Platforms (HAPs), and Unmanned Aerial Vehicles (UAVs) as airborne mediums to offer services to users in locations such as airplanes, mountainous regions, and the sea, where traditional ground-based base stations cannot provide coverage [1,2]. The use of diverse types of satellites and HAPs is expected to foster

connectivity in various industries, including scenarios involving natural disasters, seaborne base stations, and construction sites [3].

Various scenarios utilizing NTN have been presented. Among them, an air-to-ground system has been proposed, enabling communication with airborne objects by utilizing ground networks while also providing ground network services in the air [4]. Additionally, research is being conducted on the NB-IoT (Narrowband Internet of Things) that supports IoT applications within NTN, as well as eMTC (enhanced Machine Type Communication) [5]. Over time, NTN is evolving to offer communication services or an expanded range of communication services to the public based on 5G technology, and numerous studies are currently underway in this domain. Covered also are the development of 5G NTN standards and the challenges of satellite 5G network integration technology and an overview of the state of the art in LEO satellite access [6,7]. There are numerous factors to consider when applying NTN. It should be capable of supporting terrestrial networks and be adaptable to various industries [8]. Moreover, as technology progresses, studies are being conducted to incorporate Artificial Intelligence (AI) and Machine Learning (ML) models into communication systems [9]. Several papers have explored the application of ML models in different fields relevant to communication systems [10,11]. Additionally, research is underway on AI/ML management in communication systems, including methods that leverage data exchanged during handover processes [11–14].

Several considerations arise when high-altitude satellites or UAVs are utilized to provide services to ground terminal nodes. Among them, the handover of the control plane is crucial for ensuring service continuity when a User Equipment (UE) moves from one cell served by a base station to a neighboring cell, or transitions from a receiving base station to a neighboring base station's cell [15,16]. However, applying the handover metric used in terrestrial networks to Non-Terrestrial-Network (NTN) poses a challenge. In a Terrestrial Network (TN), the distance between the UE and the base station is relatively close, allowing handover indicators to be measured at relatively short distances and used for triggering handovers. In contrast, in the NTN system, satellites can be positioned at altitudes ranging from 2000 km to 35,786 km above sea level, requiring considerably longer distances than in TN [17]. This difference in distance poses a significant challenge in developing handover mechanisms and metrics that are suitable for NTN's unique characteristics.

Furthermore, satellites provide services while moving at a constant speed in the Earth's orbit. Due to this constant motion, the User Equipment (UE) receiving service from such a satellite experiences frequent handovers, even if it remains stationary. This frequent handover leads to a degradation in communication service quality and poses challenges in ensuring continuous service quality. Additionally, when a high-altitude satellite provides service to a ground-based UE, a propagation delay occurs. In the case of Measurement-Based Handover Triggering, the propagation delay is based on the measurement result obtained from the reference signal at the satellite's location. However, there exists a disparity between the measurement result obtained when the satellite is located and the result obtained at the reference point after accounting for the actual propagation delay. This discrepancy adds complexity to the handover process and necessitates careful consideration to mitigate its impact on communication system performance. Because of reasons, several papers have presented diverse approaches to address the NTN handover problem [18–22]. These include proposing a graph framework and determining an optimal path, optimizing the handover process through MIMO (Multiple Input Multiple Output) technology, maximizing benefits by defining a utility function for mobile terminals, detecting load during the handover process and employing channel reservation optimization strategies, as well as introducing papers focusing on optimizing handover by enhancing the random access process.

To address the challenges in servicing ground terminal nodes at high altitudes, this paper considers the realistic scenario, taking into account the propagation delay that occurs at high altitudes, and proposes a handover triggering technique based on machine learning to predict the distance and altitude angle between the moving satellite's cell center and the

User Equipment (UE). The XGBOOST model, an ensemble algorithm based on gradient boosting, is utilized to predict the distance and elevation angle between the UE and the cell center after the propagation delay based on the UE's location. Using a model that determines whether to trigger handover based on the predicted location can enhance the handover function to ensure continuous service provision at high altitudes and achieve more accurate handover triggering. Section 2 of this paper provides a comprehensive review of the related literature, including performance measurement of NTN handover and evaluations of proposed handover triggering techniques based on mobility enhancement by 3GPP. Section 3 describes the proposed system model, the dataset used for experimentation, and the implementation methodology. Section 4 elaborates on the performance evaluation metrics used to assess the research results and presents the outcomes of the model's performance. Finally, Section 5 offers a concluding summary of the paper.

## 2. Related Work

### 2.1. Literature Review

Experiments were conducted in several papers to evaluate 5G-based handover performance in NTN. The performance of terrestrial and non-terrestrial networks was compared using conditional handover. Most of the studies were conducted based on CHO [23–25]. Handover performance was measured in the FR2 band frequency band. The performance measurement comparison target was compared with the terrestrial network, and the KPIs used are shown in Table 1 [23].

**Table 1.** Key Performance Indicators (KPIs) of Conditional Handover.

| KPIs |
| --- |
| HO Success |
| All Mobility Failures |
| Ping Pong |

HO Success means whether the UE succeeded in handover to the target cell, All Mobility Failures means whether HOF occurred from the source cell to the RLF or target cell, and Ping Pong (PP) means whether or not the UE returned to the source cell within 1 s from the last handover. Table 2 shows the parameters used in the experiment [23].

**Table 2.** FR2 band handover experiment parameter.

| Experiment Parameter | Handover Type | Preparation Offset (db) | Execution Offset (db) | UE Speed (km/h) | Cell Count |
| --- | --- | --- | --- | --- | --- |
| Variable | BHO, CHO | 3, 7 | 3, 6 | 3, 30, 60, | 1, 4 |

In paper [23], $o_{prep}$ means execution when the source cell differs from the target cell by the corresponding value, and $o_{exec}$ means execution when the target cell differs from the source cell by this value. As a result of the experiment, the CHO HO Success value was superior to BHO when the UE's moving speed was 30 km/h or higher. It was confirmed that the higher the $o_{prep}$ value, the better the HO Success. However, when a large $o_{exec}$ value was given, the HO Success value was low due to the late handover execution and the PP significantly decreased. If the $o_{exec}$ offset is large, performing handover execution in a situation where the target cell signal strength is worse than the currently serving cell will lead to a handover, which lowers the handover success rate. Conversely, when the $o_{exec}$ value is relatively small, the signal strength of the target cell is good, which increases the handover success rate. However, the closer the handover is to the target cell, the better the signal strength of several nearby cells, which increases the probability of PP.

In addition, a CHO-based UE-oriented handover scheme has been proposed. There will be too many resource requests to handle the handover of all UEs in the satellite, and a technique for handling the process in the UE has been proposed to distribute it. In addition, the handover between several candidate satellites rather than one satellite was dealt with, and the experiment was conducted by dividing the number of candidate satellites, elevation angle, signal strength, and service time-based handover [24]. Among the proposed methods for mobility enhancement in 3GPP Rel in 16, research has also been conducted using the location-based handover triggering technique. The cell radius is defined as $R_c$ and $R_c-$ and $R_c+$ are defined as conditions for location-based handover triggering (LHT) to occur, limiting handover triggering in the range not included in the corresponding section and, at the same time, satellite service according to time. A method using the rate of change of the distance from the center of each cell was also proposed. Table 3 illustrates the parameters used in the experiment [25].

**Table 3.** Handover experiment parameter.

| Experiment Parameter | Handover Type | Triggering Type |
|---|---|---|
| Variable | BHO, CHO | MHT, LHT(R), LHT(P), LHT(R,P) |

The paper assumed HOM as 0 db and TTT as 0 ms. When comparing the performance of Measurement-Based Handover Triggering (MHT) and Location-Based Handover Triggering (LHT), it was confirmed that LHT performed better.

In order to solve the problems existing in NTN with ML, methods applied in 3GPP Rel in 18 and several papers are proposed [9–14]. There are papers applying machine learning (ML) to improve performance in NTN. In [10], the ML techniques applied to each situation in NTN are summarized. For example, Reinforcement Learning (RL) Q-Learning is used to improve moving cell connectivity, and trajectory optimization [11], and Mobile Edge Computing (MEC) and RL are used to efficiently provide services to unserved areas. Application example, RL example for throughput increase, RL for secondary link backup, Bidirectional forwarding detection (BFD), Long term-short memory (LSTM) and Neural Network (NN) application example, multi-layer for disaster relief Perceptron (MLP)—Examples of applying LSTM and Deep Q-Learning for Broadcasting/Multicasting are summarized. Most of the problems given in NTN showed an approach to solve with RL by defining the target to be optimized as an object function [26–29]. In addition, examples and architecture of MEC offloading for user data traffic in NTN were shown [29,30].

### 2.2. Non-Terrestrial Network Overview

Existing terrestrial 5G networks provide communication services around installed base stations. Telecommunications companies want to provide services with wide coverage at low cost. However, in order to cover all domestic terrestrial networks using terrestrial networks, terrestrial cell coverage must be installed without any vacancy. NTN provides a wide range of coverage at high altitudes and can provide communication services in mountainous areas where ground networks are not installed, in the middle of the sea, and in airplanes. In order to apply NTN with these advantages to the communication system, 3GPP listed and suggested solutions to problems for providing NTN services from an overall overview of NTN through Releases 15, 16, and 17. In order to actually service and commercialize using NTN, several papers have been proposed to solve the problems presented by 3GPP. In this section, representative contents of NTN are summarized.

In 3GPP Release 16, scenarios defined by dividing GEO and LEO into transparent payload and regenerative payload were described. Each scenario provided by NTN is shown in Figures 1 and 2 [31].

There are several definitions of satellites according to altitude. Table 4 is a representative satellite.

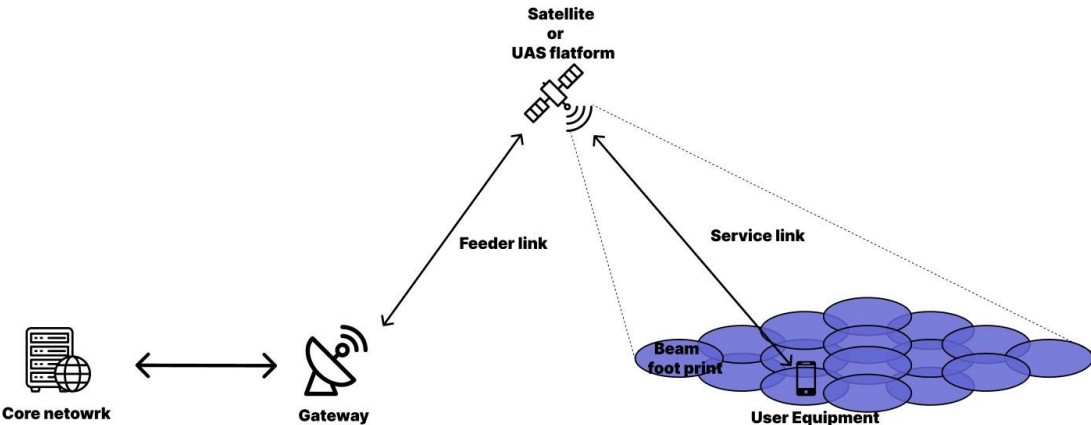

**Figure 1.** Transparent payload in 3GPP Release 16.

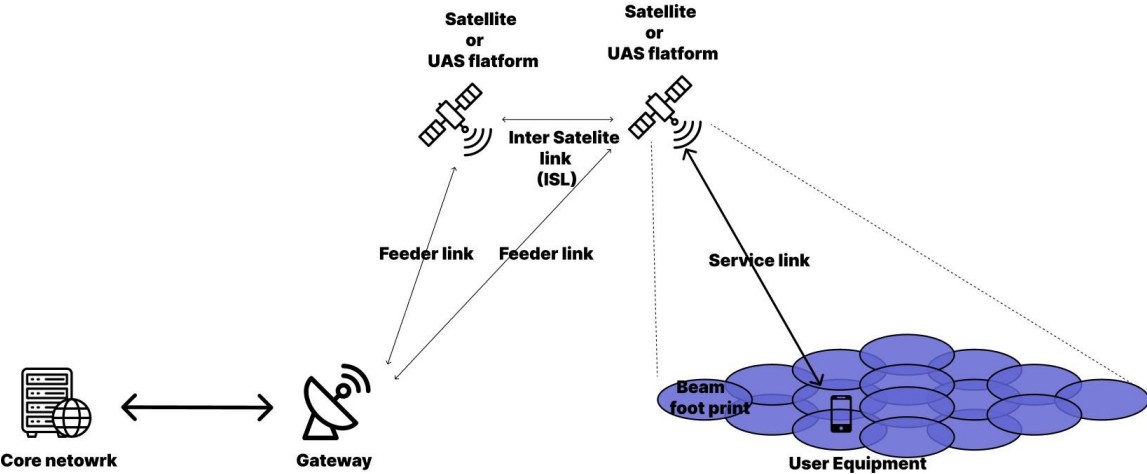

**Figure 2.** Regenerative payload in 3GPP Release 16.

**Table 4.** Definition of satellite in 3GPP Rel 16.

| Platforms | Altitude Range (km) | Orbit | Beam Foot Print Size (km) |
| --- | --- | --- | --- |
| Low-Earth Orbit (LEO) satellite | 300–1500 | Circular around the earth | 100–1000 |
| Medium-Earth Orbit (MEO) satellite | 7000–25,000 | Circular around the earth | 100–1000 |
| Geostationary-Earth Orbit (GEO) satellite | 35,786 | Notional station keeping position fixed | 200–3500 |

Several NTN network elements exist, and the interface between each network element is shown in Table 5.

**Table 5.** Definition of interface type between network elements.

| Network Elements | Interface Type |
| --- | --- |
| UE—Satellite | NR Uu |
| Satellite—Satellite | Inter Satellite Link (ISL) |
| Satellite—gNB | NR Uu |
| gNB—5G Core(5GC) | NG |
| Satellite(gNB on board)—Gateway | Satellite Radio Interface (SRI) |
| Satellite(gNB on board)—5GC | NG |
| Satellite(gNB-DU on board)—Gateway | Satellite Radio Interface (SRI) |
| Satellite(gnB-DU on board)—gNB-CU | F1 |
| gNB-CU—5GC | NG |

Table 6 is an NR-RAN-based NTN architecture proposed based on the interface of Table 5.

**Table 6.** Types of NTN architectures.

| NTN Architecture |
| --- |
| Transparent satellite |
| Regenerative satellite—gNB processed payload |
| Multi-connectivity involving transparent NTN-based NG-RAN and cellular NG-RAN |
| Multi-connectivity between two transparent NTN-based NG-RAN |
| Regenerative satellite—gNB-DU processed payload |
| Multi-connectivity involving regenerative NTN-based NG-RAN (gNB-DU) and cellular NG-RAN |
| Multi-connectivity between two generative NTN-based NG-RAN (gNB on board) |

The scenario presented in TR38.821 of 3GPP Rel 16 provides services at high altitudes. Thus, there is a delay issue according to distance. Table 7 is a table showing overall values corresponding to the scenarios defined in Table 4.

**Table 7.** Characteristics of scenarios.

| Scenario | Geo-Based NTN | Leo-Based NTN |
| --- | --- | --- |
| Orbit Type | Notional station keeping position fixed | circular orbiting around the earth |
| Altitude (km) | 35,786 | 600/1200 |
| Payload | Transparent/Regenerative | |
| Max beam footprint size (km) | 3500 | 1000 |
| Max Round Trip Delay (ms) (Transparent/Regenerative) | 541.46/270.73 | 41.77/20.89 |

Unlike the terrestrial network, NTN propagation delays for each scenario in Table 7 cause delays in the measurement report, handover command reception, and handover request/ack in the handover process. In order to trigger handover, a measurement report is transmitted to the receiving cell or gNB when various measured values satisfy the criteria. However, the NTN environment that communicates at high altitudes cannot trust the validity of the measured values. Figure 3 shows the difference in signal attenuation received at the cell center and cell edge in terrestrial and non-terrestrial networks.

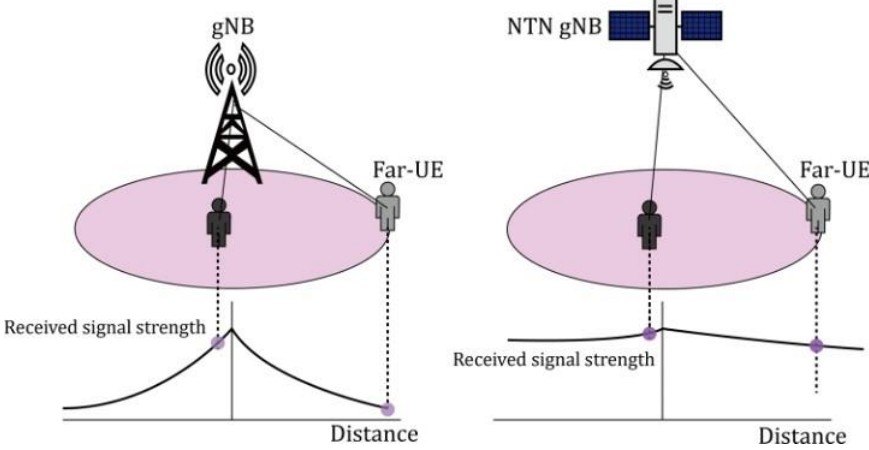

**Figure 3.** Difference in received signal attenuation between TN and NTN.

In the left image depicting the terrestrial network, there is a noticeable difference in signal strength between User Equipments (UEs) located close to the gNB (ground-based station) and those located farther away. However, in the right image representing the

Non-Terrestrial Network, the signal strength difference between the UE at the cell center of the NTN gNB and the UE at the cell edge is not clearly evident. Therefore, using signal strength as a handover triggering method in NTN may lead to inaccuracies. To address this issue, the corresponding TR 38.821 suggests various handover triggering methods available in NTN, which go beyond relying solely on signal reception sensitivity. Table 8 presents the proposed handover triggering methods in NTN.

**Table 8.** Method of handover triggering enhancement.

| Triggering Method |
| --- |
| Measurement-based triggering |
| Location-based triggering |
| Time/Timer-based triggering |
| Timing advance value-based triggering |
| Elevation angles of source and target cell-based triggering |

TR 36.763 describes the IoT NTN scenario. Among the scenarios of 3GPP TR 38.821 Rel 16, a transparent payload scenario is assumed. The difference from the scenario of 3GPP TR 38.821 Rel 16 is that the Medium Earth Orbit (MEO) scenario is added. MEO is a satellite located between the LEO altitude range and the GEO altitude range [32]. IoT NTN must be able to support EPC and 5GC. And it is assumed that the UE can support Global Navigations Satellite System (GNSS). Table 9 summarizes the contents of the delay issue in LEO and GEO. IoT NTN considers the MEO scenario. Therefore, Table 9 summarizes the MEO delay and related parameters.

**Table 9.** Characteristic of MEO satellite.

| Scenario | MEO-Based Non-Terrestrial Access Network |
| --- | --- |
| Orbit type | Circular orbiting at medium altitude around the earth |
| Altitude (km) | 10,000 |
| Payload | Transparent |
| Max beam footprint size (km) | 4018 |
| Max Round Trip delay (ms) | 186.9 (service and feeder links) |

loT NTN assumes NB-IoT and eMTC support. In the case of NB-IoT, if it is out of cell coverage provided by NTN, it is treated as Radio Link Failure and the UE attempts RRC connection for re-establishment. In the case of eTMC, there are issues such as delay for handover signaling processing, measurement validity, frequent handover, dynamic neighbor cell list, handover of a large number of UE, and impact of propagation delay difference in measurements such as NR NTN.

*2.3. Handover*

The terminal node is connected to the base station that has expanded the cell site at the current location and uses the communication service. Connected terminal nodes may stay in one location. However, there are cases where it continues to move. In this case, the location may be moved from a source cell to an adjacent cell, or the adjacent cell may be a cell of a base station other than a cell expanded from one base station. Handover occurs when moving from one cell to an adjacent cell. Handover is a process to ensure continuity of service. Figure 4 shows a case in which handover occurs as the user moves to an adjacent cell as the user moves.

There are several criteria for handover triggering. Representative A1–A6 events among them are shown in Table 10.

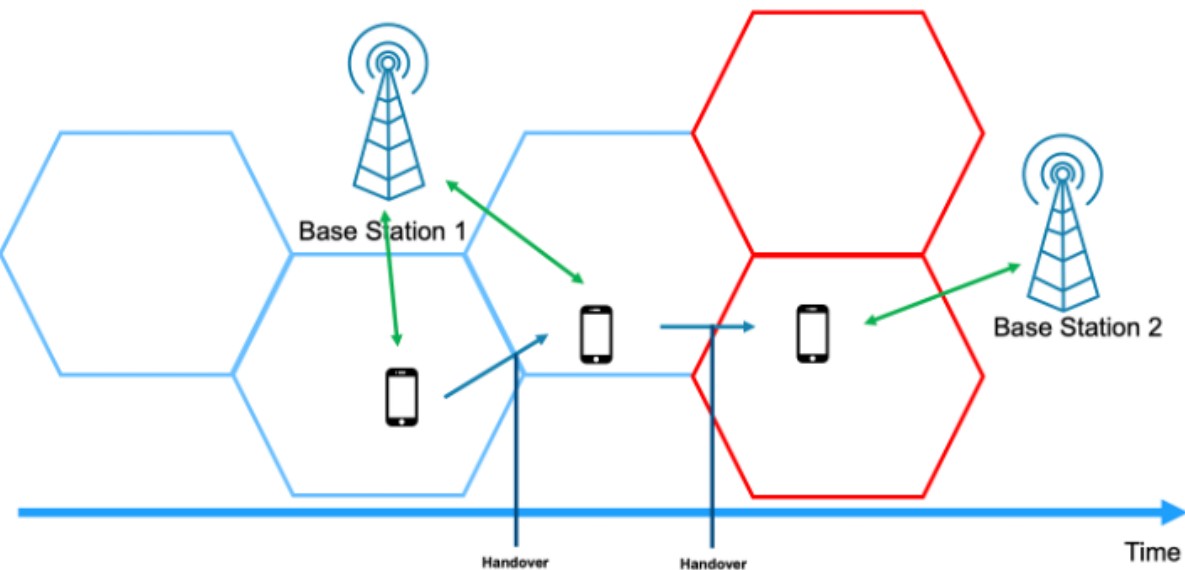

**Figure 4.** Handover scenario.

**Table 10.** Definition of handover triggering A events.

| Event Type | Description |
| --- | --- |
| A1 | Serving becomes better than threshold |
| A2 | Serving becomes worse than threshold |
| A3 | Neighbor becomes amount of offset better than PCell/PSCell |
| A4 | Neighbor becomes better than threshold |
| A5 | PCell/PSCell becomes worse than absolute threshold1 AND Neighbor/SCell becomes better than another absolute threshold2 |
| A6 | Neighbor becomes amount of offset better than SCell |

Based on the events in the table above, handover is performed from the receiving cell to the adjacent cell. Efficient handover minimizes dropped calls or loss of data transmission, and mobility management allows users to move seamlessly while maintaining a reliable connection.

Handover works differently in different situations. There are several cases such as handover between cells within one base station, handover between base stations, and handover from 5GC to EPC. An inter-cell handover process within one of the base stations will be described. Conditional Handover (CHO) is a handover that improves the handover process performance of Baseline Handover (BHO). Handover process triggering can be determined by using additional criteria. In addition, configuration information of an adjacent cell list capable of handover other than RACH to the target cell after the RRC reconfiguration step is stored in the UE. Then, if the conditions for handover triggering are satisfied, the handover process is performed on the target cell. Therefore, CHO can ensure more robust connectivity and continuity of service. Figure 5 shows the flow of CHO.

The UE measures the criterion of the measurement index. If the standard of the measurement index is met, a measurement report is transmitted to the source cell. CHO is requested for target cells that can handover the source cell. The candidate target cell list transmits a handover request ack message to the source cell. Then, the source cell transmits an RRC Reconfiguration message including the configuration of CHO target cells to the UE. The UE transmits an RRC Reconfiguration Complete message to the source cell. After receiving the CHO configuration, the UE maintains a connection with the source

cell and determines CHO execution conditions for the target cell list. Once a target cell from the target cell list satisfies the condition, the UE detaches from the source cell, applies the configuration of the cell that meets the condition, proceeds with synchronization with the target cell, and transmits the RRC Reconfiguration Complete message to complete the RRC handover procedure. The UE releases the stored CHO configuration after successful completion of the RRC handover procedure.

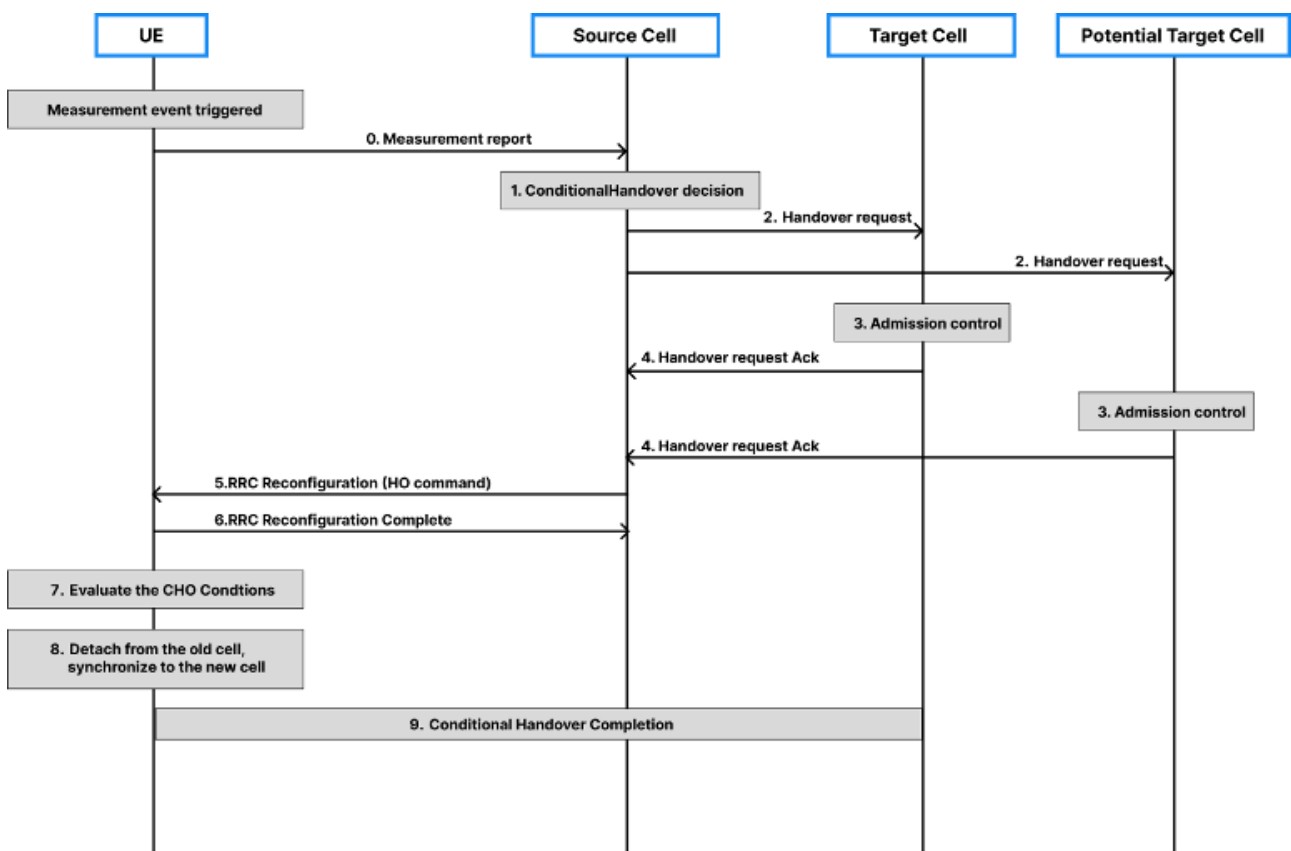

**Figure 5.** Flow of conditional handover.

## 3. System Model

### 3.1. The Proposed Model

In the NTN system, in order for the UE to receive the location information of fast-moving satellites at high altitudes such as LEO and MEO, the data transmitted by the reference signal from the satellite to the UE at the time the UE receives it is already old data and has an error as much as the propagation delay. Papers [13,14] applying the ML algorithm presented a method to solve the LEO-based handover problem. In paper [13], a handover process prediction model for a terminal approaching a cell boundary was proposed through feature preprocessing using a K-means clustering algorithm. In paper [14], RSRP values and handover flags of several beams serviced by satellites were formulated and the values of the formulas were obtained for each 0.5 s time period. Paper [14] also proposed a model that predicts handover from the current time zone to the next beam using a CNN model. For satellites serving at high altitudes, the difference in signal strength between the center of a cell and the edge of a cell is not clear due to the long communication distance. In addition, MEO satellites as well as LEO satellites require service at higher altitudes. This means that the propagation delay between the satellite and the UE is different depending on the altitude. The previously reviewed papers presented a handover triggering model that did not consider propagation delay and the results [23–27].

In order to improve these points, this paper proposes a handover triggering model that uses the distance and altitude angle between the center of the cell and the UE instead of using the difference in indistinct signal strength and considers the propagation delay according to the altitude. The proposed model considers the propagation delay in the suggested range when the satellite transmits data to the UE in order to consider it close to the actual situation when the NGSO satellite is deployed in the NTN. According to 3GPP in Release 16, LEO and MEO satellites have different propagation delays according to their respective altitudes. The experiment is conducted considering the corresponding range and additionally the propagation delay at higher altitudes. As a feature for predicting handover triggering, handover triggering is predicted after propagation delay using the distance and elevation between the cell center and the UE. Among the NTN scenarios proposed by 3GPP in Release 16, a GNSS system is supported by assuming a CHO-based transparent payload satellite, the UE can use the GNSS system, and the satellite used for NTN can broadcast cell information. The assumption is also made that the channel is not time-varying and is in a Line of Sight (LOS) condition. Figure 6 is the handover triggering flow chart of the proposed model.

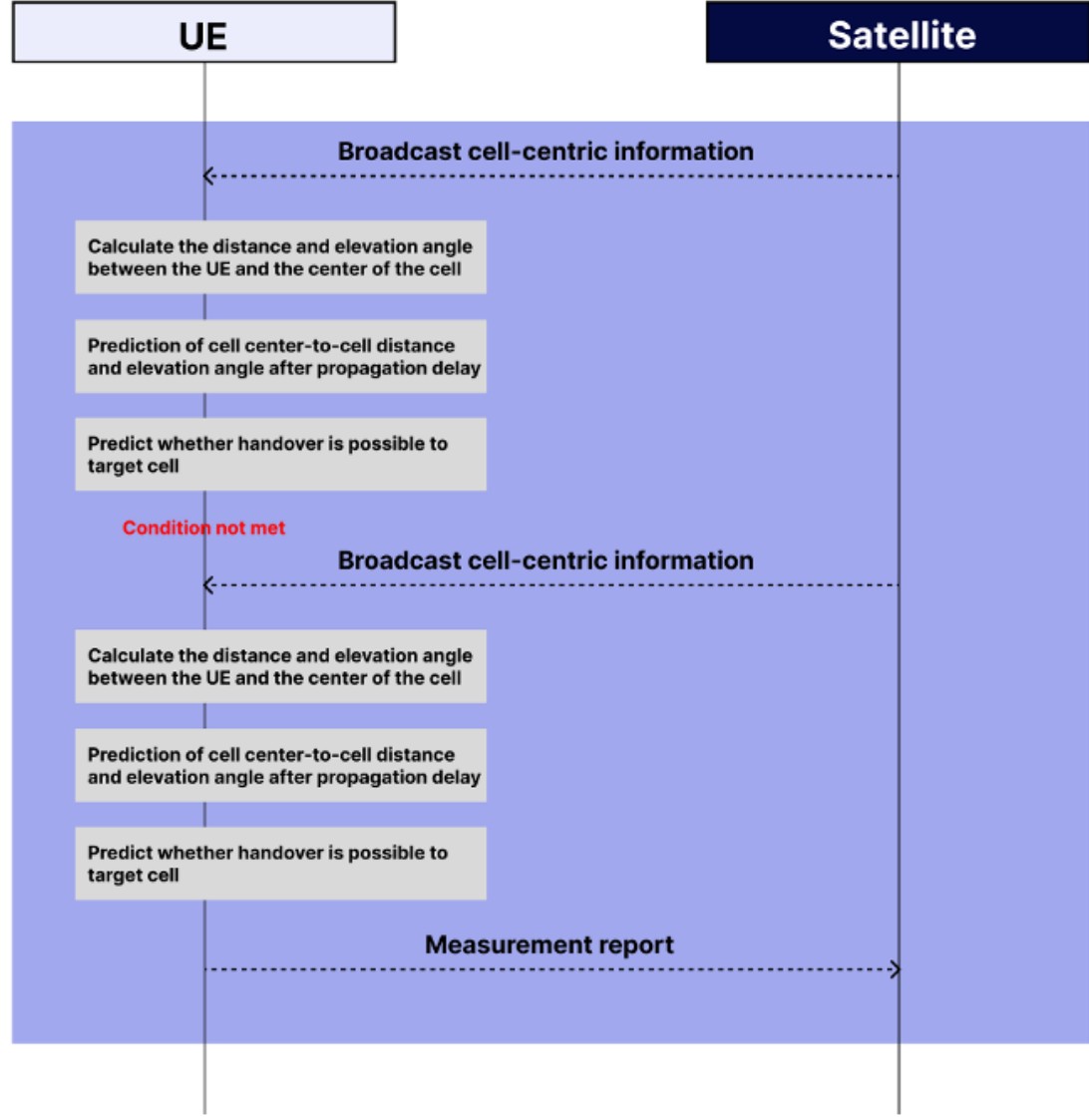

**Figure 6.** Flow chart of handover triggering.

Upon receiving the location information of the satellite, the UE calculates the center distance and elevation angle between cells based on the location information. Then, the calculated value is obtained through a model that predicts the distance and altitude angle after the propagation delay. Based on that value, whether to generate a measurement report is predicted through a model that predicts handover triggering. Figure 7 shows the system model for learning.

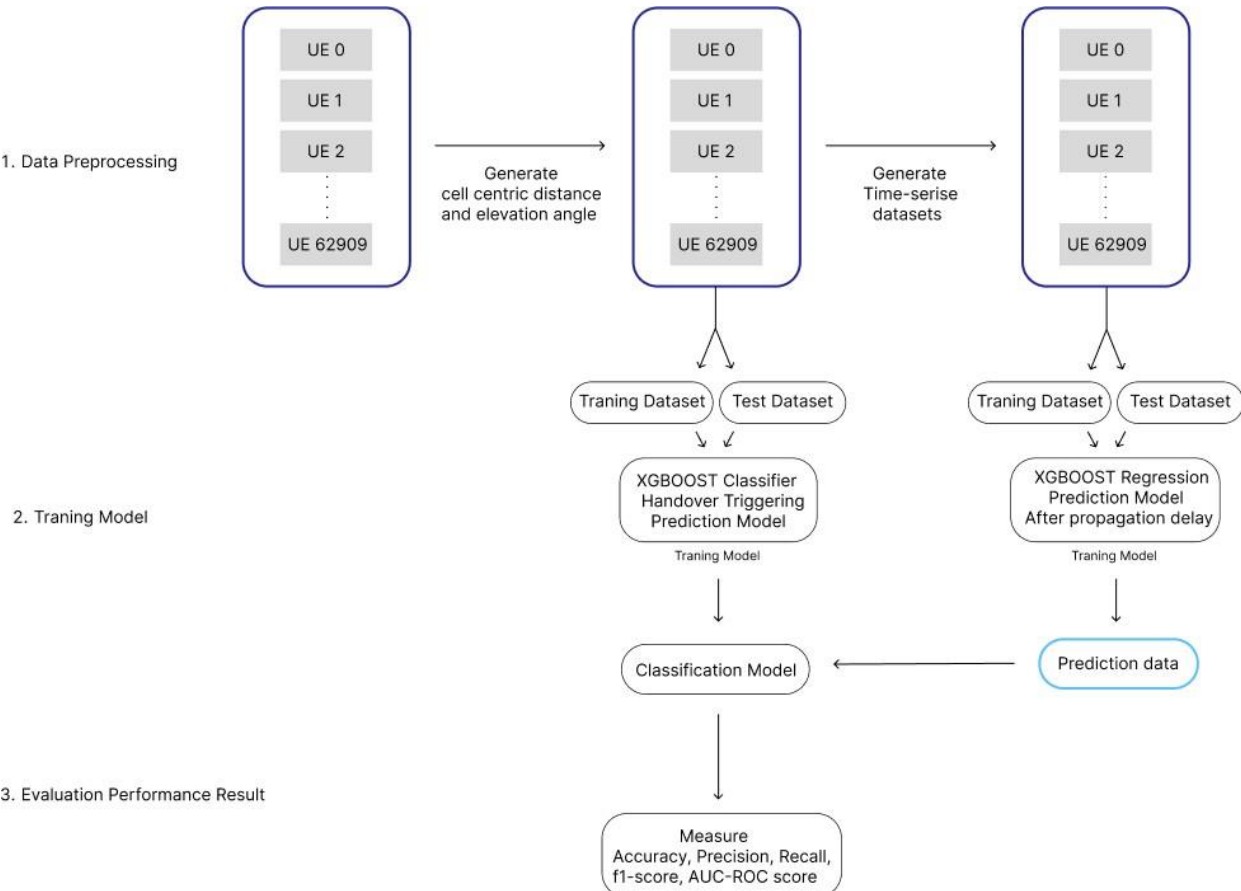

**Figure 7.** Flow chart of system model.

### 3.2. Dataset

In the data set, one satellite serves three cells as a beam. The data set consists of two parts. The location information data set of the UE and the satellite serving three cells are cell-centered latitude and longitude data at intervals of 0.001 s according to the movement of the satellite. To collect the two data sets, a simulator using the STK 11.3 engine between the satellite and the UE was created. The STK engine is composed of a C#-based net framework. The objects used in the simulator are satellites and several UEs. About 60,000 UEs were randomly distributed around three cells. If there is a UE within the coverage among the three cells serviced by the satellite, the UE and the satellite are connected through a service link and the corresponding data is collected.

After creating two data sets, the distance between the center of each cell and the UE from the satellite presented in Section 3.3.1 in this paper and the altitude angle between the UE from the center of the satellite presented in Section 3.3.2 were created. The diameter of the cell is 250 km, cell number 0 in the middle is the cell that the UE is receiving service from, and the cells on the left and right are candidates for the target cell. In conclusion, a simulation data set was used in which one moving satellite serves three cells. Figure 8 shows a satellite supporting three cells.

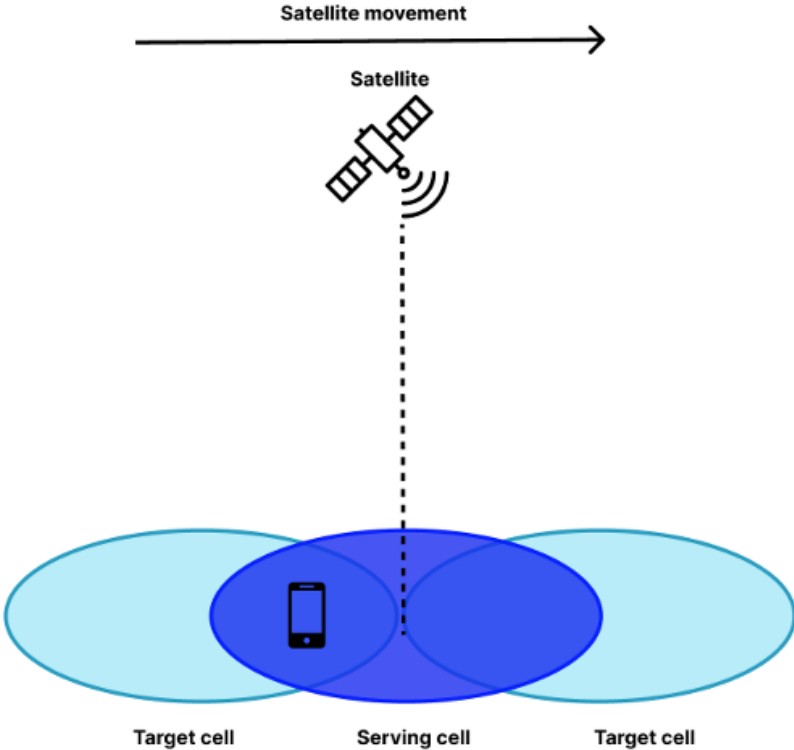

**Figure 8.** Moving satellite simulation supporting three cells.

There are location data sets centered on target cells and serving cells according to satellite motion per 0.001 ms, and data sets containing UE location information. The characteristics of each data set are shown in Tables 11 and 12.

**Table 11.** Feature description of cell-centric dataset.

| Feature | Description |
| --- | --- |
| Time (UTGC) | The time at which the cell center of the satellite was measured |
| Latitude (deg) | Latitude of cell center |
| Longitude (deg) | Longitude of cell center |
| Altitude (km) | Altitude of the satellite |
| Latitude Rate (deg/s) | Cell center latitude change rate |
| Longitude Rate (deg/s) | Cell center longitudinal change rate |
| Altitude Rate (km/s) | Satellite altitude change rate |

**Table 12.** UE dataset features used.

| Feature | Description |
| --- | --- |
| Cell type | Composed of Serving Cell, Target Cell, Candidate cell |
| Latitude (deg) | Latitude of cell center |
| Longitude (deg) | Longitude of cell center |
| Altitude (km) | Altitude of satellites |
| Time (s) | The time when UE is serviced by cell |

Dataset Distribution Analysis

The overall distribution of the range of features used to train the model was analyzed and visualized. Figure 9 shows the change in the distribution of the altitude angle and the distance between cell centers of a satellite at the location of one UE during 600 s of simulation.

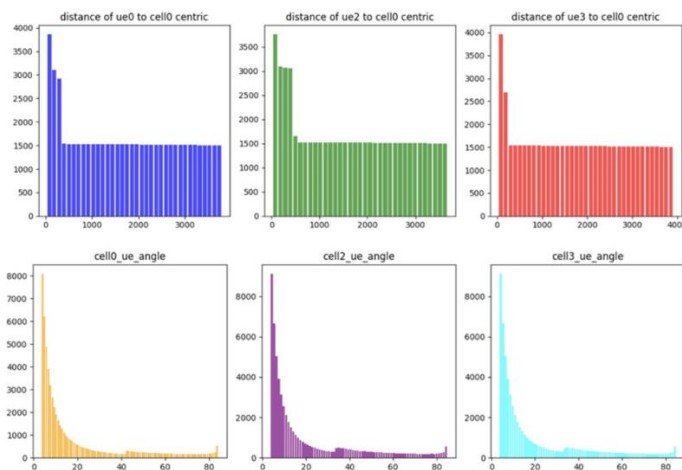

**Figure 9.** Changes in feature distribution during simulation.

As the UE moves away from the satellite, the distance between each cell increases, and it can be confirmed that most of the distance between the satellite and the UE is long. In addition, if the altitude angle between the satellite and the UE is large, it means that the satellite and the UE are adjacent, and it can be confirmed that most of them are similarly far away. Figure 10 confirmed the distribution of the min and max values of the distance between the centers of each cell and the elevation angle in the 3000 UE data.

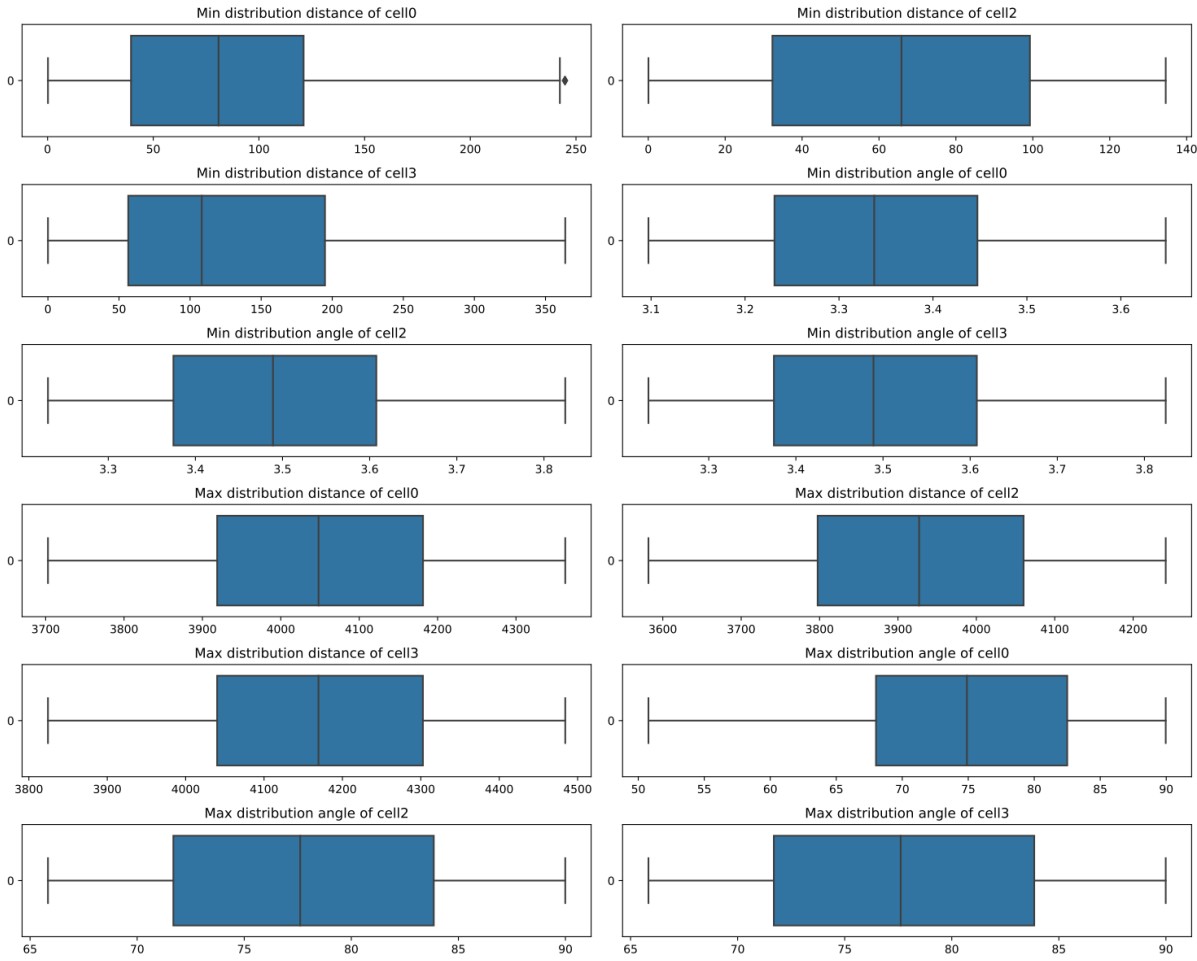

**Figure 10.** Min, Max distribution of 3000 UEs.

Most of the distribution of the distance between the UE and the cell center and the elevation angle formed a constant distribution.

### 3.3. Data Preprocessing

#### 3.3.1. Distance between UE and Cell Centric

A cell-centered data set over time is a data set that represents the 0.001 ms change of a feature. Using the cell location information in the data set and the location information in the UE data set, the distance and elevation angle between the UE and the cell center per 0.001 s were calculated. The haversine formula was used to calculate the distance between the cell center and the UE on Earth. This mathematical formula is cited from paper [13].

$$\theta = \frac{d}{r} \qquad (1)$$

$\theta$ the central angle between the two points, $d$ is the distance between the two points, and $r$ is the radius of the sphere.

$$hav(\theta) = hav(\varphi_2 - \varphi_1) + \cos\varphi_1 \cos\varphi_2 hav(\lambda_2 - \lambda_1) \qquad (2)$$

$hav(\theta)$ can be calculated directly from the latitude ($\varphi_2$, $\varphi_1$) and longitude ($\lambda_2$, $\lambda_1$) of the two points.

$$hav(\theta) = sin^2\left(\frac{\theta}{2}\right) = \frac{1 - \cos\theta}{2} \qquad (3)$$

Above is the haversine function applied to the central angle, latitude, and longitude. To get the distance ($d$) from this function, you need to use the arc-haversine or arcsin function.

$$d = r \cdot archav(h) = 2r \cdot \arcsin\left(\sqrt{h}\right) \qquad (4)$$

The formula applied with the above *archav* or arcsin is solved and expressed as follows.

$$d = 2r\arcsin\left(\sqrt{hav(\varphi_2 - \varphi_1) + (1 - hav(\varphi_2 - \varphi_1) - hav(\varphi_2 + \varphi_1)) \cdot hav(\lambda_1 - \lambda_2)}\right) \qquad (5)$$

$$= 2r\arcsin\left(\sqrt{sin^2\left(\frac{\varphi_2 - \varphi_1}{2}\right) + \left(1 - sin^2\left(\frac{\varphi_2 - \varphi_1}{2}\right) - sin^2\left(\frac{\varphi_2 + \varphi_1}{2}\right)\right) \cdot sin^2\left(\frac{\lambda_2 - \lambda_1}{2}\right)}\right) \qquad (6)$$

$$2r\arcsin\left(\sqrt{sin^2\left(\frac{\varphi_2 - \varphi_1}{2}\right) + cos(\varphi_1) \cdot cos(\varphi_2) \cdot sin^2\left(\frac{\lambda_2 - \lambda_1}{2}\right)}\right) \qquad (7)$$

#### 3.3.2. Elevation Angle between UE and Cell Centric

As a handover triggering method proposed for mobility enhancement in 3GPP in Rel 16, a method of utilizing the elevation angle between the cell center and the UE is proposed. Assuming that the distance between two points is d and the altitudes of the two points are *alt*1 and *alt*2, the formula for calculating the altitude angle between the two points is as follows.

$$Elevation\ angle = \arctan\left(\frac{alt_1 - alt_2}{d}\right) \qquad (8)$$

#### 3.3.3. Sampling Data

Based on the above formula, the center distance and elevation angle values between the UE and each cell were obtained, and standard scaling was used to standardize the distribution of features. There are data on the rate of change of the center distance and altitude angle between cells for each UE at 1 ms intervals. Sampling was performed with data per 10 ms to fit the propagation delay unit of NTN. In addition, N windows of past propagation delay intervals are used to predict the center distance and altitude angle between the UE and the cell after the propagation delay based on the location information

that the UE has received from the satellite. Based on the window data for each of the past N propagation delays, the center distance and altitude angle between the UE and the cell after the propagation delay are predicted. Figure 11 shows the data sampling process.

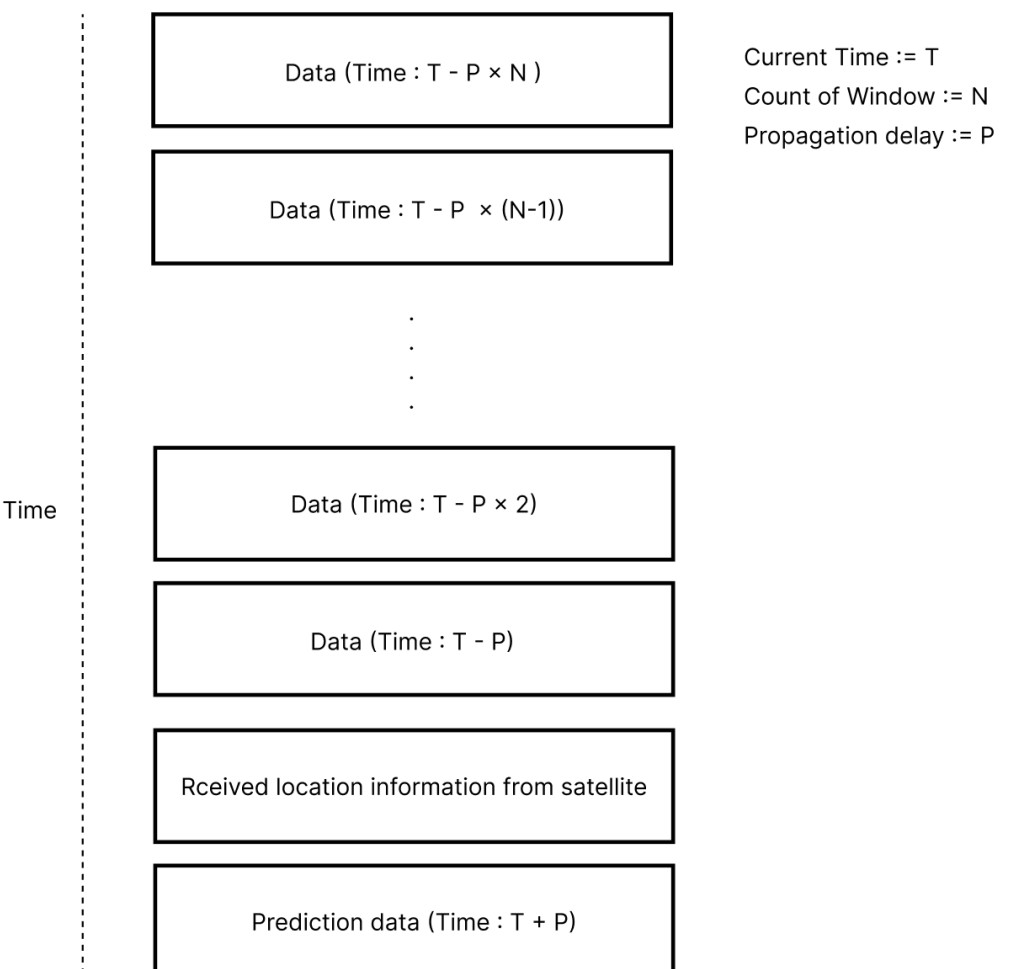

**Figure 11.** Sampling datasets.

*3.4. Implementation*

The proposed system model is divided into two steps. One is a model that predicts the distance and altitude angle between the UE and the center of the cell after propagation delay based on the location information transmitted from the satellite. One is a model that predicts handover triggering occurrence based on the predicted value after propagation delay.

3.4.1. UE-Cell Center Distance and Elevation Angle Prediction Model Considering Propagation Delay

To train the model, the data generated and preprocessed in Section 3.3.3 was used. The XGBOOST algorithm was used and regression values were predicted using XGBRegressor, and MultiOutputRegressor was used to predict multiple features. To improve the performance of the model, Bayesian optimization, which can efficiently find hyperparameters, was used. In Bayesian optimization, gamma, learning rate, max depth, estimator, and subsample were used as parameters for XGBOOST hyperparameter tuning. Each range used is shown in Table 13.

In order to compare the performance of the hyperparameter range, cross validation was performed three times using cross validation. In order to find the interval that maximizes the performance, the target of the object function was used as an indicator of neg mean squared error.

**Table 13.** Hyperparameters used for Bayesian optimization.

| Hyper Parameter | Range |
| --- | --- |
| Gamma | (0, 1) |
| Learning rate | (0.01, 1) |
| Max depth | (5, 20) |
| Number of estimator | (100, 1000) |
| Subsample | (0.5, 1) |

3.4.2. Handover Triggering Model Using UE-Cell Center Distance and Elevation Angle

In the case of the handover triggering prediction model, the model was trained by labeling the overlapping area of the cells on both sides as a region where handover is possible based on the cell center distance. The classification was performed using XGBClassifier, and hyperparameter values were adjusted using hyperopt. Table 14 shows the variables and ranges for XGBOOST hyperparameter tuning.

**Table 14.** Hyperparameters used in hyperopt.

| Hyper Parameter | Range |
| --- | --- |
| Colsample bytree | (0, 1) |
| Min child weight | (0.01, 1) |
| Max depth | (5, 20) |
| Learning rate | (100, 1000) |

## 4. Performance Results

*4.1. Performance Indicator*

4.1.1. Confusion Matrix

A confusion matrix is a matrix used in classification problems as an index to indicate the result of classification. It consists of a total of four cases: two cases as a result predicted by the model, and two cases of the actual correct answer label. Table 15 is an example of a confusion matrix.

**Table 15.** Confusion matrix.

| | Positive (Model) | Negative (Model) |
| --- | --- | --- |
| Positive (Answer) | True Positive (TP) | False Negative (FN) |
| Negative (Answer) | False Positive (FP) | True Negative (TN) |

A True Positive is when the result predicted by the model is Positive and the actual correct answer is also Positive; a False Negative is when the result predicted by the model is Negative and the actual correct answer is Positive; a False Positive is when the result predicted by the model is Positive and the actual correct answer is Negative; a True Negative is when the result predicted by the model is Negative and the actual correct answer is also Negative.

4.1.2. Accuracy Score

The accuracy score is a numerical value representing accuracy in classification. The formula for calculating the accuracy score is as follows.

It represents the actual accuracy predicted by the model and represents the proportion of true positives and true negatives in the entire data set. In the case of the accuracy score, if the inconsistency of the data set distribution is too great, it can actually be a meaningless number. Therefore, it is necessary to match the distribution of data set labels.

$$Accuracy = \frac{TP + TN}{TP + FN + FP + TN} \tag{9}$$

### 4.1.3. Precision Score

Precision represents the percentage of data that are actually true among the data that the model classifies as positive. The formula for calculating the precision score is as follows.

$$Precision = \frac{TP}{TP + FP} \tag{10}$$

### 4.1.4. Recall Score

The recall score represents the proportion of data that the model predicted to be true among data that were actually positive. The formula for calculating the recall score is as follows.

$$Recall = \frac{TP}{TP + FN} \tag{11}$$

A model that always predicts only true positives will have a recall value close to 1. Therefore, the precision score should be compared with the recall score.

### 4.1.5. F1 Score

The *F*1 score is a numerical value representing the harmonic average of the Precision score and the Recall score. It is used as an indicator to express the two performance indicators in a balanced way, and the formula is as follows.

$$F1 = 2 \cdot \frac{Precision \cdot Recall}{Precision + Recall} \tag{12}$$

It is an indicator that uses precision score and recall score together, which are in a trade-off relationship, and the closer the F1 score is to 1, the better the model performance can be evaluated.

### 4.1.6. AUC-ROC Curve

The Receiver Operation Characteristic (ROC) curve is a ratio graph showing True Positive Rate (*TPR*) versus False Positive Rate (*FPR*). The graph is represented by *FPR* on the X-axis and *TPR* on the y-axis, and the formula for obtaining each value is as follows.

$$TPR = \frac{TP}{TP + FN} = Recall = Sensitiviy \tag{13}$$

$$FPR = \frac{FP}{FP + TN} = 1 - Specificity \tag{14}$$

TPR is the percentage of correct answers that the model actually gave, and FRP is the percentage of correct answers that were not actually correct. The closer TPR is to 1 and the closer *FPR* is to 0, the better the model. The AUC score is the area of the area under the graph drawn by the ROC curve. As the model shows good performance, *TPR* approaches 1 and *FPR* approaches 0, so the graph is drawn in the upper left corner. Then, the area of the base of the AUC score also approaches 1, which is an indicator of good performance.

### 4.1.7. Mean Squared Error (MSE)

The Mean Squared Error (*MSE*) is the average of squared errors. Error is the difference between the value predicted by the model and the actual correct answer. The better the model guesses the correct answer, the smaller the *MSE* value will be. In other words, the smaller the *MSE* value, the better the performance of the model. The formula for calculating the *MSE* is as follows.

$$MSE = \frac{1}{2}\Sigma_i(y_i - \hat{y}_i)^2 \tag{15}$$

The 1/2 in the formula is to eliminate this by multiplying by 1/2 because the exponent that plays a square role when the *MSE* is differentiated is multiplied as a constant 2 in

the entire formula. $y_i$, $\hat{y}_i$ means the correct answer predicted by the model and the actual correct answer, respectively. As the *MSE* is a quadratic function, it can be seen that the value of the loss function changes greatly as the error increases. When using *MSE* as the loss function, the model adjusts its parameters in the direction that minimizes the error between the given data and the predicted values. By iterating through the aforementioned process, the model seeks to find the optimal parameters.

### 4.2. Performance Result

Figure 12 shows the MSE predicted by the cell center and elevation angle XGBOOST regression prediction model for each propagation delay.

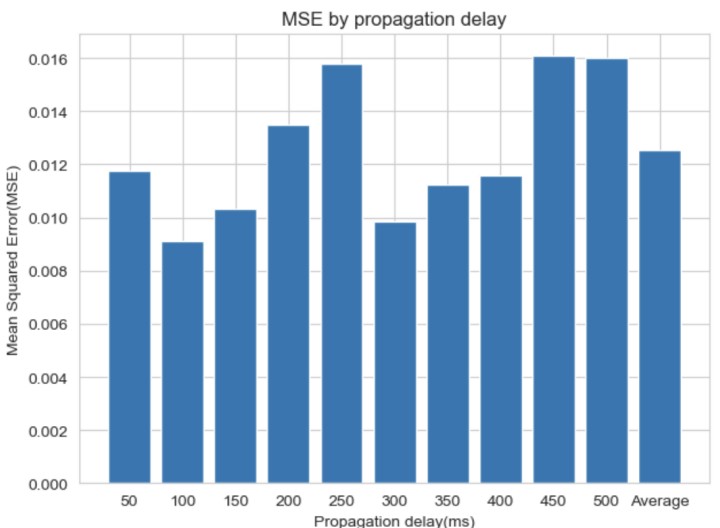

**Figure 12.** MSE by propagation delay.

As a result of the evaluation, the performance was about 0.01252 on average when measured by MSE, which is a regression performance index. Figures 13–17 show the results of using the results predicted by the regression prediction model as test data for the handover triggering prediction model using the cell center and elevation angle. Figure 18 is a graph showing the improved performance as a percentage when each model is applied.

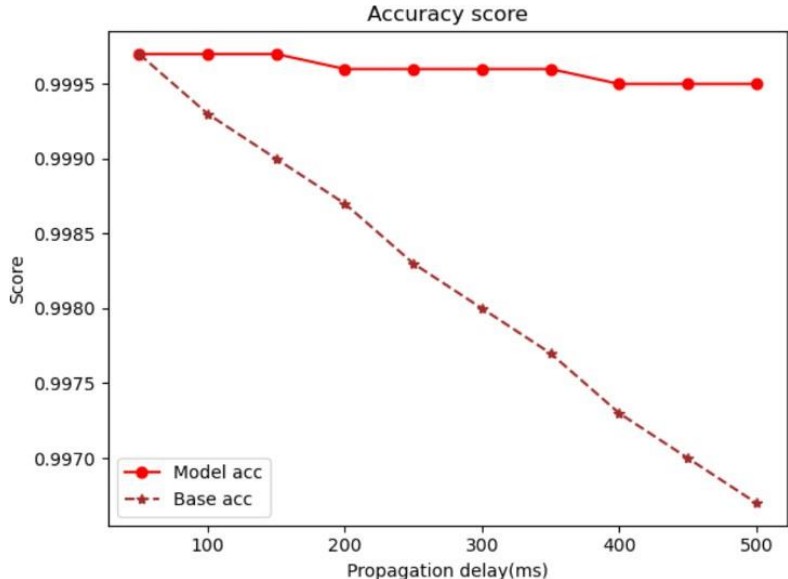

**Figure 13.** Accuracy score.

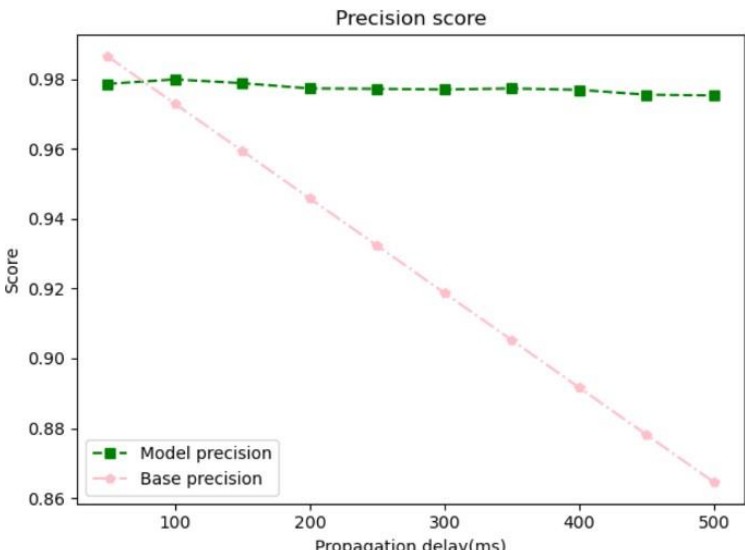

**Figure 14.** Precision score.

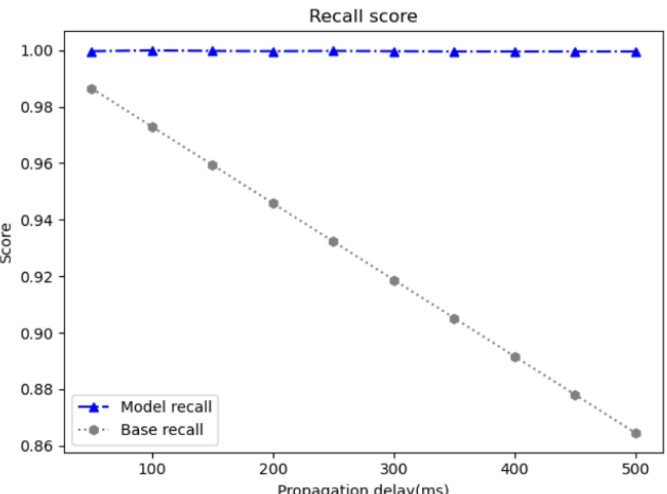

**Figure 15.** Recall score.

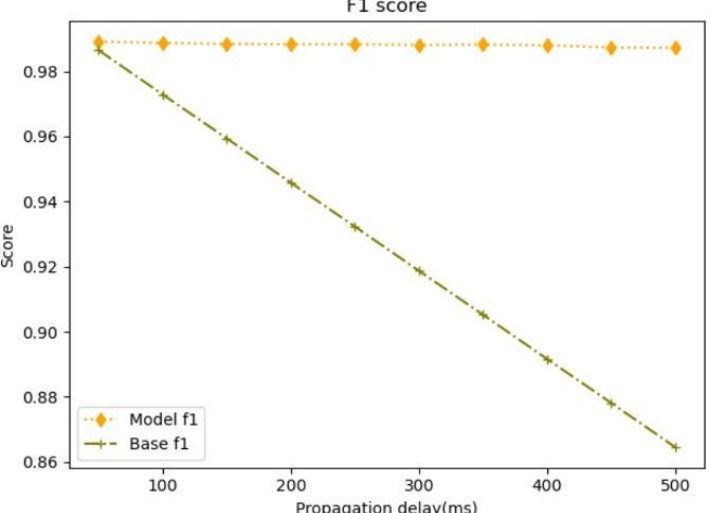

**Figure 16.** F1 score.

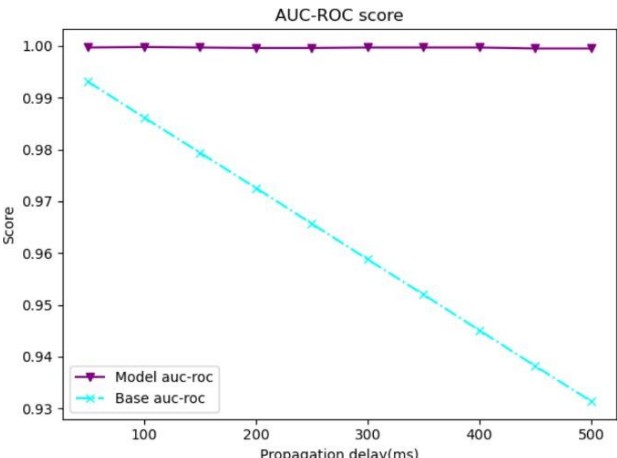

**Figure 17.** AUC-ROC score.

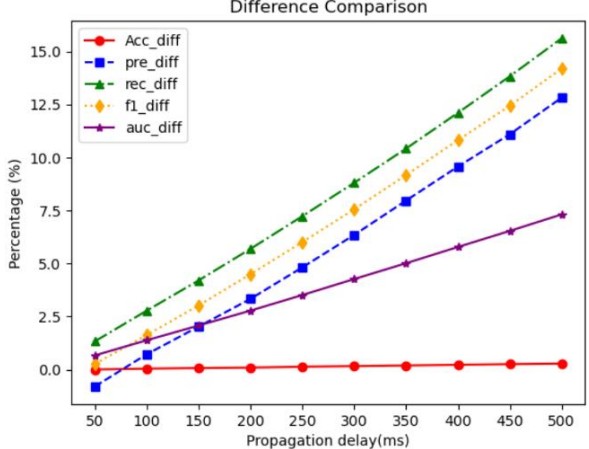

**Figure 18.** Improved Performance Comparison.

Among the graphs showing the results, the recall and precision scores are indicators that evaluate the degree to which handover triggering occurs well at the time when it should occur, and that it occurs at the time when it should not occur. In the case of recall, this can consist of Radio Link Failure or Handover Failure, and in the case of precision, it can lead to Unnecessary Handover (UHO) or PP. In the experiment, 40 out of 50 UEs were used as training data, and 10 UEs were used as test data. Each UE was simulated for 600 s. Figure 19 shows the RLF + HOF, UHO + PP of the handover triggering test results considering the propagation delay.

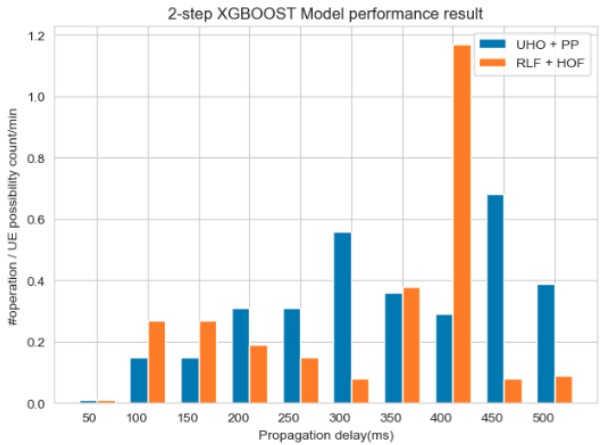

**Figure 19.** Two-step XGBOOST performance results.

The result of Figure 19 is a graph showing the point at which the UE can predict handover triggering for 1 min on average when the model makes predictions by considering the propagation delay. When conducting the experiment in paper [29], HOM was set to 0 db and TTT to 0 ms, and the test was conducted. This means that it is a graph that measures the result of performing a measurement report immediately when the condition of handover triggering is satisfied. Therefore, in the case of the prediction model using LHT + EHT, the graph is shown based on whether the prediction of the first point where handover triggering is possible is consistent with the assumption that HOM operates like 0 db and TTT operates like 0 ms. When the two-step XGBOOST model is applied, it shows good performance in all propagation delay error intervals. Even when applied to the actual NTN environment, this represents a more accurate prediction than handover triggering, which does not consider the existing propagation delay for the section of the moving satellite. In particular, the recall score indicates a score that predicts the point in time when handover triggering should occur, and it can be seen that the prediction succeeds in most cases. This model shows strength in situations that can lead to RLF and HOF, which are used as KPIs for simulations in several papers.

## 5. Conclusions

The purpose of providing communication services through satellites is to extend coverage to areas that cannot be reached by terrestrial networks. As a hyper-connected era is anticipated in the future, Non-Terrestrial Network (NTN) must ensure continuous and reliable communication in all regions above a certain altitude. While there are various challenges in applying NTN, the issue of mobility in connected mode is of utmost importance. Users may experience significant inconvenience if communication services are interrupted, emphasizing the critical need for uninterrupted connectivity. Satellites at high altitudes provide services to ground terminal nodes, but due to the long distance, there is a time delay for radio waves to reach the terminals. This temporal difference becomes more significant at higher altitudes. Moreover, Low Earth Orbit (LEO) satellites, operating at relatively low altitudes, move at high speeds compared to ground-based terminals, posing a challenge to maintaining communication continuity. Communication quality is bound to deteriorate even when satellites operate at higher altitudes. Our study conducted experiments in an environment closer to real-world communication situations by considering propagation delay, an actual environmental variable, rather than assuming only ideal conditions. In an actual NTN environment, Line-of-Sight (LOS) conditions may not always prevail, introducing more unpredictable variables. Obstacles between communication terminals, unexpected orbit deviations, and changes in channel quality over time are likely occurrences. To address this, this paper proposed a two-step XGBOOST model that compensates for propagation delay. The two-step XGBOOST model demonstrates superior performance across most altitude ranges compared to the existing handover triggering method based on New Radio (NR) that does not consider propagation delay, ensuring communication continuity with moving satellites. The handover triggering experiment considering propagation delay yielded favorable performance, especially from the 50 ms section. Although there is a slight difference in precision, the f1-score, which represents the overall model performance, shows better results. Thus, the proposed model exhibits enhanced handover triggering performance not only in LEO NTN and Medium Earth Orbit (MEO) NTN but also across the entire altitude range.

**Author Contributions:** Conceptualization, E.K.; Methodology, E.K.; Software, E.K.; Validation, E.K.; Investigation, E.K.; Data curation, E.K.; Writing—original draft, E.K.; Writing—review & editing, E.K.; Visualization, E.K.; Supervision, I.J.; Project administration, I.J.; Funding acquisition, I.J. All authors have read and agreed to the published version of the manuscript.

**Funding:** This work was supported by the Institute of Information & Communications Technology Planning and Evaluation (IITP) grant funded by the Korean government (MSIT) (No. 2020-0-00107, Development of the technology to automate the recommendations for big data analytic models that define data characteristics and problems).

**Data Availability Statement:** The data that support the findings of this study are available on request from the corresponding author, I.J. The data are not publicly available due to their containing information that could compromise the privacy of research participants.

**Conflicts of Interest:** The authors declare no conflict of interest.

## Abbreviations

The following abbreviations are used in this manuscript:

| | |
|---|---|
| 3GPP | 3rd Generation Partnership Project |
| 5G | 5th Generation |
| BHO | Baseline Handover |
| CART | Classification And Regression Tree |
| CHO | Conditional Handover |
| EPC | Evolved Packet Core |
| GEO | Geostationary Earth Orbit |
| GNSS | Global Navigations Satellite System |
| HAP | High-Altitude Platform |
| HOF | Handover Failure |
| HOM | Handover Margin |
| ISL | Inter Satellite Link |
| LEO | Low Earth Orbit |
| MEO | Medium Earth Orbit |
| NB-IoT | Narrowband-Internet of Things |
| NGSO | Non-Geostationary Orbit |
| NTN | Non-Terrestrial Network |
| O-RAN | Open Radio Access Network |
| PCell | Primary Cell |
| PP | Ping Pong |
| PSCell | Primary-Secondary Cell |
| RACH | Random Access Channel |
| RRC | Radio Resource Control |
| SCell | Secondary Cell |
| SRI | Satellite Radio Interface |
| TN | Terrestrial Networks |
| TTT | Time To Trigger |
| UAV | Unmanned Aerial Vehicle |
| UE | User Equipment |
| XGBOOST | Extreme Gradient Boosting |
| eMTC | Enhanced Machine-Type Communication |
| gNB | Next Generation Node B |
| MIMO | Multiple Input Multiple Output |

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
