# Peer review of "Handover Triggering Prediction with the Two-Step XGBOOST Ensemble Algorithm for Conditional Handover in Non-Terrestrial Networks"

_electronics, doi:10.3390/electronics12163435_

Round 1

Reviewer 1 Report

This article proposes a 2-Step XGBOOST ensemble algorithm to predict handover triggering in Non-Terrestrial Networks (NTN), which can predict when user equipment (UE) needs to switch to a new network and consider propagation delay to improve network performance. The innovation lies in the 2-Step XGBOOST model that incorporates propagation delay and satellite motion into prediction. The first step predicts distance and elevation angle post-propagation delay when satellite position information is relayed to the UE. The second step utilizes this data to predict handover triggering. The model demonstrates superior performance in maintaining communication continuity and quality. In my view, this work falls within the scope of Electronics and should be published after making some modifications. A few suggestions and clarifications that need to be addressed before publication are presented below.

1. The introduction is quite lengthy and touches upon multiple subjects. It is suggested to break it down into several paragraphs, each focusing on a single subject, to enhance readability.

2. On page 4, concerning the sentence “However, when a large Oexec value was given, the HO Success value was low due to the late handover execution, but the PP significantly decreased”, it would be beneficial to provide an explanation on how the conclusion of “the PP significantly decreased” was derived, as it cannot be inferred from the graph. Furthermore, on page 18, the paper mentions “As the MSE is a quadratic function, it can be seen that the value of the loss function changes greatly as the error increases”. The relationship between MSE and the loss function is mentioned, but not explained in detail. Could more detailed information be provided?

3. It is suggested to replace the term “receiving sensitivity” with “signal attenuation”. The term “receiving sensitivity”, which refers to the minimum signal strength that a receiver can detect, is mentioned multiple times in Figure 5 and its description. To compare the receiving sensitivities of two receivers, it is necessary to know at what level of signal strength the receiver starts to fail in reliably receiving signals. However, what is shown in Figure 5 is the change in received strength with distance, thus using the term “receiving sensitivity” is not accurate.

4. The paper employs many abbreviations, such as “5GS”, “EPC”, “RACH”, “RRC”, and “CDHO”, but does not provide clear explanations or definitions for these abbreviations. This may confuse readers, especially those unfamiliar with the technical terminology used in this field. It is recommended to provide additional explanations.

5. On page 17, in section 4.1.1, there is an error in the sentence “False Negative when the result predicted by the model is Positive and the actual correct answer is Negative”. According to Table 15, “False Negative” should be corrected to “False Positive”.

Author Response

Dear reviewer,

Thank you for taking you precious time for reviewing paper.

I attach a document that answer of your comments and suggestions.

Thank you.

Reviewer 2 Report

This paper proposed a model that can predict the distance and elevation angle between the UE and the center of the cell considering the propagation delay. This reviewer has the following concern:

1. It is natural for UE to use satellite ephemeris data or relevant information to perform the prediction. Is the proposed method better than the prediction method based on ephemeris data? Are there any experimental comparisons?

2. It is suggested to cite the following two papers and explain the innovation of this work compared with these literature.

[R1] Dahouda, Mwamba Kasongo, Sihwa Jin, and Inwhee Joe. "Machine Learning-Based Solutions for Handover Decisions in Non-Terrestrial Networks." Electronics 12, no. 8 (2023): 1759.

[R2] Zhang, Chenchen, Nan Zhang, Wei Cao, Kaibo Tian, and Zhen Yang. "An AI-based optimization of handover strategy in non-terrestrial networks." In 2020 ITU Kaleidoscope: Industry-Driven Digital Transformation (ITU K), pp. 1-6. IEEE, 2020.

3. Many of the formulas in this paper are exactly the same as those in [R1], so it is recommended to refer to references.

Author Response

(The authors gave the same response as above.)

Reviewer 3 Report

1.      The authors need to clarify why the NTN link is measured based on values to determine that the handoff is not valid as the counterpart of the NT links.

2.      The sentences in lines 49, 52, 80 – 84 either do not render the intended meaning, have not been written well, or contain sentence fragments; please correct those parts.

3.      What do the authors mean by the sentence in lines 58 – 59?

4.      In lines 87, 92, 93, and 94 probably, the authors need to use ‘Section 2’ instead of ‘Chapter 2.’

5.      The authors should highlight the contribution of the study.

6.      Please allow spacing after the ending sentence in line 121.

7.      Figure 3, 4, and 12 is not well visible.

8.      Some Figure caption end with a dot, whereas some Figures are not. Please make it consistent according to the journal guideline.

9.      Please use ‘km’ instead of ‘Km’ in line 266.

10.   Please describe how the used data were generated/collected.

Author Response

(The authors gave the same response as above.)

Round 2

Reviewer 2 Report

There are a number of studies that consider the use of satellite ephemeris data in handover. A few of these are listed below. The authors proposed a method to predict the distance and altitude angle after the propagation delay. My question is that the UE can utilize the ephemeris data to predict the trajectory of satellites over time [A6], then it does not seem difficult to calculate the distance and altitude angle after the propagation delay with the known trajectory.

[A1]Z. Wu, F. Jin, J. Luo, Y. Fu, J. Shan and G. Hu, "A Graph-Based Satellite Handover Framework for LEO Satellite Communication Networks," in IEEE Communications Letters, vol. 20, no. 8, pp. 1547-1550, Aug. 2016.

[A2]L. Feng, Y. Liu, L. Wu, Z. Zhang and J. Dang, "A Satellite Handover Strategy Based on MIMO Technology in LEO Satellite Networks," in IEEE Communications Letters, vol. 24, no. 7, pp. 1505-1509, July 2020.

[A3]Y. Wu, G. Hu, F. Jin and J. Zu, "A Satellite Handover Strategy Based on the Potential Game in LEO Satellite Networks," in IEEE Access, vol. 7, pp. 133641-133652, 2019.

[A4]Y. Liu, X. Tang, Y. Zhou, J. Shi, M. Qian and S. Li, "Channel Reservation based Load Aware Handover for LEO Satellite Communications," 2022 IEEE 95th Vehicular Technology Conference: (VTC2022-Spring), Helsinki, Finland, 2022, pp. 1-5.

[A5]J. Wang, W. Mu, Y. Liu, L. Guo, S. Zhang and G. Gui, "Deep Reinforcement Learning-based Satellite Handover Scheme for Satellite Communications," 2021 13th International Conference on Wireless Communications and Signal Processing (WCSP), Changsha, China, 2021, pp. 1-6

[A6] X. Lin, S. Cioni, G. Charbit, N. Chuberre, S. Hellsten and J. -F. Boutillon, "On the Path to 6G: Embracing the Next Wave of Low Earth Orbit Satellite Access," in IEEE Communications Magazine, vol. 59, no. 12, pp. 36-42, December 2021.

[A7]Wang, Zhifei, Xiangming Wen, Zhaoming Lu, Wenpeng Jing, and Yujing Zhang. "Random access optimization for initial access and seamless handover for 5G-satellite network." Computer Networks 214 (2022).

[A8]M. Hosseinian, J. P. Choi, S. -H. Chang and J. Lee, "Review of 5G NTN Standards Development and Technical Challenges for Satellite Integration With the 5G Network," in IEEE Aerospace and Electronic Systems Magazine, vol. 36, no. 8, pp. 22-31, 1 Aug. 2021.

Author Response

Dear reviewer,

Thank you for taking your valuable time for a review.

I have attached the content of your reply.

Thank you.

Reviewer 3 Report

It is hard to see the texts in Figure 13. This figure demands better visualization.

The English looks fine in the revised version.

Author Response

(The authors gave the same response as above.)

Round 3

Reviewer 2 Report

I have no further comments.